# Consumer Attitudes and Purchase Intentions toward Food Delivery Platform Services

**Han-Shen Chen** [1,2,*], **Chia-Hsing Liang** [3], **Shu-Yi Liao** [3] and **Hung-Yu Kuo** [1]

1   Department of Health Diet and Industry Management, Chung Shan Medical University, Taichung City 40201, Taiwan; jason87523@yahoo.com.tw
2   Department of Medical Management, Chung Shan Medical University Hospital, Taichung City 40201, Taiwan
3   Department of Applied Economics, National Chung Hsing University, Taichung 40227, Taiwan; 520gamm@gmail.com (C.-H.L.); sliao@nchu.edu.tw (S.-Y.L.)
*   Correspondence: allen975@csmu.edu.tw; Tel.: +886-4-2473-0022 (ext. 12225); Fax: +886-4-23248188

**Abstract:** With the advent of the Online to Offline (O2O) era, the rise of various food delivery platforms not only provides consumers with more choices, but also allows restaurant operators to reach more potential consumers and increase their additional revenue. This study is based on theory of planned behavior (TPB), and includes the 'utilitarian value' and 'hedonic value' as research variables. Structural equation modeling (SEM) was used to verify the research hypotheses, and to analyze consumers' purchase intentions toward online food delivery platforms. An online survey was also conducted, and a total of 1300 questionnaires were distributed. After excluding invalid questionnaires with incomplete answers, a total of 1082 questionnaires were deemed valid, and the effective recovery rate was 83.23%. The research results were as follows: (1) the attitude, subjective norms, and perceived behavioral control of consumers will have a significant positive effect on utilitarian value and hedonic value; (2) the utilitarian and hedonic values have a significant positive effect on purchase intention; and (3) the utilitarian and hedonic values have a mediating effect on attitude, subjective norms, perceived behavioral control, and purchase intention. Based on the above results, food delivery platform operators can identify the key factors that drive consumers to use their services in order to formulate effective management strategies and create greater business opportunities for their organizations.

**Keywords:** services marketing; utilitarian value; hedonic value; consumer behavior; sustainable business models

---

## 1. Introduction

With the advent of the sharing economy, the development of e-commerce, the promotion of third-party payment solutions, and the growth of the stay-at-home economy, food delivery platforms have emerged. Food delivery platform operators have created a new technological model for food delivery services that allow consumers to connect to various local restaurants and food providers through a mobile app. Quick search functions allow consumers to add the dishes they want to order to their shopping cart. Consumers can be immediately provided with an estimated delivery time depending on their designated location, or they can specify their desired delivery time. Then, a delivery person can accept the order, go to the restaurant to pick up the order, and deliver it to the consumer to complete the service. After placing an order, consumers can use a smart tracking function in the app to track the progress of their order. Meanwhile, the app can record the consumer's preferences. Food delivery platform operators have created a new model for food delivery services. Well-known food delivery platforms include Deliveroo in the UK, Justeat in the US, and Foodpanda in Germany.

Even Uber and Amazon have entered the food delivery market by developing Uber Eats and Amazon Restaurants.

Kapoor and Vij [1] found that because consumers can follow the instructions provided on food delivery platforms and can track order fulfillment in real-time, there is no need to communicate with restaurant staff, which reduces the amount of energy spent and the amount of interpersonal communication. Alalwan [2] found that food delivery platforms make ordering food more attractive by reducing the perception of waiting time. It also helps to reduce the expensive call charges incurred as consumers call the store to ask about the status of their order, which helps reduce transaction costs. He et al. [3] found that the participation of third-party food delivery service providers is beneficial to the parties involved. For restaurants, they can add a new source of income without increasing the number of seats or staff. For consumers, they have more choices when they want to order meals, and they can access relevant information such as comments and reviews about restaurants. In addition, food delivery platforms can generate a stable commission-based income using this business model. According to the National Restaurant Association (NRA), 20% of consumers prefer to order online rather than conduct face-to-face transactions with restaurant employees. In addition, 42% will choose restaurants that offer online ordering services over those that do not [4].

The theory of planned behavior (TPB) is used to explain and predict the behavior patterns of consumers in specific circumstances. Since its development, TPB has been widely used to explore personal behavior in different research domains, such as health, environmental protection, and consumer behavior [5–14]. Mouakket [15] used TPB to study the intentions of university students to continue using Facebook in the United Arab Emirates. Kim et al. [16] used the extended TPB to predict the behavioral intention of the users of social networking sites. Ho et al. [17] used the TPB and personality traits to analyze and compare factors that cause addiction in the form of excessive social networking use among adolescents and adults in Singapore. Roos and Hahn [18] used the theory to analyze the potential value and belief structure of consumers regarding collaborative consumption. Hwang and Kim [19] used it to explore consumers' views on the green value of food delivery services via unmanned aerial vehicles (UAV). Quevedo-Silva et al. [20] applied it to explore consumers' intentions to purchase food through the internet. Upon reviewing past research involving the use of the TPB, it can be said that the theory has been widely used across several fields, and has yielded significant results. In this study, the TPB was used to explore the effect of factors such as attitude, subjective norms, and perceived behavioral control on the purchase intentions of food delivery platform users.

The shopping values that receive attention from consumers include the utilitarian value and hedonic value, which affect their satisfaction and behavioral intentions [21]. Previous studies have shown that utilitarian and hedonic shopping values are important predictors of online shopping behavior [22–27]. Babin et al. [28] argued that utilitarian value comes from consumers' shopping experience, which is why purchase decisions can be made more subjectively. Yeo et al. [29] perceived that hedonic value has a significant effect on the behavioral intention to use food delivery services. Ozturk et al. [30] pointed out that hedonic and utilitarian values affect consumers' continued use of mobile devices to book hotels. Therefore, utilitarian value and hedonic value are included in this study.

Based on the above, this study is based on TBP. TPB has been used to discuss lots of field, e.g., new services, technology and issues, which is the reason why this study uses TPB. According to the TPB, when customers use online food platform services, the attitude which means customers will think about the online food delivery platform may cause benefits and losses at the same time; as such, they will estimate the value from the online food delivery service before they use it, so the attitude of customers is very important. Subjective norms mean that the impact from other people will affect the customers' purchase intention, and it is very easy to get feedback information from the internet, so subjective norms are a key factor which affect consumers' purchase intentions. The perceived behavioral control is also a key factor which affects consumers' purchase intentions, because when consumers think it is not easy to use or it is difficult to find what they want, they would rather buy food by themselves than use an online food delivery platform. Jeng [31] used TPB to discuss Virtual Reality

(VR) services; Lung-Guang [32] discussed artificial intelligence, and their research model included TPB theory; Fleming et al. [33] used TPB to discuss the issue of community pharmacists' beliefs regarding engaging patients about prescription drug misuse. Yarimoglu and Gunay [34] used the extended TPB to examine Turkish customers' intentions to visit green hotels. Kautish et al. [35] used TPB to research consumers' environmental consciousness and recycling intentions on green purchase behavior.

Here, utilitarian value and hedonic value are used to explore the perceptions that Taiwanese consumers have towards the services offered by food delivery platforms, and the factors that affect their purchase decision. Utilitarian value and hedonic value are the immediate benefits for consumers, and that is the most important advantage of online food delivery compared to the offline. There are few studies that use TPB, utilitarian value and hedonic value at the same time to discuss food delivery platforms, but it is necessary to put utilitarian value and hedonic value into this study. First, the ease of use and the efficiency of making a deal is very important; this is why the platforms came into being, and it is the most different from a traditional food delivery service. Second, the hedonic value is an additional value of online food delivery platforms. If the purchase experience is good, it will stimulate consumers' use the platform next when they want to get a good meal. Yeo et al. [29] used hedonic value to discuss the intention of using online food delivery services, but didn't use utilitarian value; Avcilar and Özsoy [36] used perceived utilitarian value and hedonic value to discuss online shopping intentions, but this study didn't use the theory of planned behavior to find other factors that may affect consumers' purchasing intentions. Pahnila and Warsta [37] used utilitarian value, hedonic value, and social factors and habits to discuss the factors that may affect consumers' purchasing intentions in online shopping. Khare and Rakesh [38] found that consumers' purchases through online shopping websites are be influenced by utilitarian value, the attitude toward online shopping, the availability of information, and hedonic values, but do not know whether the subjective norms or perceived behavioral control will affect them. As such, the results of this study will help clarify behavioral patterns surrounding the use of food delivery platforms, which were identified in past studies, but have not yet been tested.

## 2. Literature Review

### 2.1. Purchase Intention

Purchase intention refers to the probability of consumers to purchase products. The higher the likelihood, the stronger the purchase intention [39,40]. Understanding consumers' purchase intentions can help companies to analyze the market and adjust their products or services in ways that increase sales and generate more profits [41]. Schlosser et al. [42] pointed out that consumers' trust in websites directly and positively affects their purchase intention. Chakraborty [43] pointed out that the brand awareness and perceived value will ultimately influence the purchase intentions of consumers. Chang and Chen [44] confirmed that web site quality and web site brands affect consumers' trust and perceived risk, and in turn, consumer purchase intention. Thamizhvanan and Xavier [45] pointed out that prior online purchase experience and online trust have significant impacts on customers' purchase intentions. Ariffin et al. [46] pointed out that consumers' perceived risks have a significant negative influence on consumer's online purchase intentions, especially the social risk. In addition, consumers' purchase intentions are affected by information from third parties, such as experts, scholars, and news media [47]. Based on the previous study, purchase intentions are necessary to discuss with regard to online food delivery platforms, and it is the ultimate purpose that the platform manager wants for the consumer. Additionally, we know that there are lots of factors that may impact a consumer, so this study will clarify part of them.

### 2.2. Shopping Value

Shopping value is widely valued in the domain of consumer behavior; while many consumers value service quality or product quality, shopping value has been found to be more important [48,49].

Jones et al. [25] argued that the customer experience would generate shopping value for customers, which includes the utilitarian value obtained from products and services, along with the value experienced on an emotional level. This kind of thinking, where shopping value is extended into the shopping process, presents multi-faceted factors, namely, utilitarian value and hedonic value [21,46,50]. Utilitarian value can refer to whether consumers can efficiently obtain the value of various products or services. Hedonic value refers to whether consumers are satisfied during the shopping process [51]. Ryu et al. [52] argued that utilitarian and hedonic value are the basis of consumers' evaluations of their consumption experience. Babin and Krey [53] argued that personal shopping value includes utilitarian value and hedonic value. Additionally, Rajan [54] pointed out that the shopping motive factors of hedonic and utilitarian have been utilized to draw conclusions about the various effects of online shopping behavior. Therefore, consumers' values can be more wholly understood with these two concepts. Utilitarian value and hedonic value are used as facets of the shopping value derived from the use of food delivery platforms in this study. These values are two important factors that platform managers need to know, because they are the immediate perceptions of consumers after using online food delivery platforms. These values will be described below.

### 2.2.1. Utilitarian Value

Jones et al. [25] defined utilitarian value as the acquisition of products and services in an efficient manner, wherein the shopping process is treated as a task, and can be viewed as reflecting a cognitive, goal-oriented behavior. Crowley et al. [55] analyzed 24 product categories; they found that "useful," "beneficial," "wise," and "valuable" were variables of utilitarian emotion. To et al. [26] found that cost–savings, convenience, varied choices, information availability, lack of social contact, and customized goods or services are important factors that influence utilitarian value in online shopping. Liu et al. [56] used utilitarian and hedonic satisfaction to clarify the overall satisfaction of online shopping. Through the above factors, customers evaluate whether their shopping experience meets their expectations, and then purchase the products or services they require. Yeo et al. [29] found that the utilitarian value of online food delivery services will significantly affect consumers' willingness to use said service. Ray et al. [57] noted that ease of use would significantly affect consumers' intentions to use online food delivery platforms.

Anderson et al. [58] pointed out that consumers' utilitarian values have a significant positive effect on their purchase intentions when shopping online. Nejati and Moghaddam [59] found that both hedonic value and utilitarian value are important factors in determining consumers' intention to go to restaurants. Lin et al. [60] shows that consumers' utilitarian and hedonic values have a significant effect on the search intention of online shopping, and then cause the shopping behavior. Ashraf et al. [61] revealed that personal innovativeness, face consciousness and uniqueness have positive effects on continuance intention in social media. Teng and Wu [62] argued that utilitarian and hedonic shopping goals influence shoppers' commitment to online shopping. Overby and Lee's study [50] showed that utilitarian value and hedonic value have positive effects on online shopping intentions. Mittal et al. [63] pointed out that perceived usefulness significantly predicts university students' use intentions on mobile applications. Based on results from earlier studies, we realized that the consumers' utilitarian value is very important and necessary to put into the research framework. The following was hypothesized:

**Hypothesis 1 (H1).** *A consumer's utilitarian value has a significant positive effect on their purchase intentions on online food delivery platforms.*

### 2.2.2. Hedonic Value

Hirschman and Holbrook's study [64] showed that hedonic value reflects the shopping value obtained from the sensory, imaginative, and emotional experience of shopping. Its focus is not only on obtaining products or completing tasks, but also on the pleasure generated while shopping. Therefore,

compared to utilitarian value, hedonic value is more subjective and individual, and more value can be obtained from fun and interest.

Dedeoglu et al. [65] argued that hedonic value lies in the value components of entertainment and emotion in products or services. Batra and Ahtola [66] noted that hedonic emotions could be classified as pleasant, nice, happy, and agreeable. Crowley et al. [55] defined nice, happy, agreeable, and pleasant as the dimensions of hedonic value. Voss et al. [67] listed twelve dimensions of hedonic value, which include fun, excited, delightful, thrilling, and enjoyable. After using online shopping platforms, customers can generate hedonic value by evaluating whether the process produced pleasure and fun, whether it helped them forget the stress they were experiencing, and the amount of time they spent immersed in the shopping experience [25,68,69]. Alalwan [2] indicated that hedonic value has a crucial effect on the satisfaction derived from e-commerce services, as well as the continued intention to use them. Yeo et al. [29] found that hedonic value has a significant effect on purchase intention toward online food delivery. Ozkara et al. [70] found that hedonic value has a significant positive effect on consumers' intentions toward online shopping. Childers et al. [22] also found that consumers' hedonic value is one of the most important factors with regard to shopping intentions online. To et al. [26] pointed out that hedonic value has positive effect on search intentions, and further impact on online shopping intentions. Chiu et al. [49] argued that both utilitarian value and hedonic value are positively associated with consumers' repeat purchase intentions. Wang [71] argued that hedonic values have higher positive associations with customers' intentions to buy than with the intent to search information. Ali et al. [72] pointed out that hedonist value has an effect on consumers' purchase intentions. Wen-Kuo et al. [73] pointed out that hedonic value is the main influence on impulse purchasing. Based on results from earlier studies, hedonic value has a very significant effect on consumers' behavior online. Therefore, hypothesis H2 is investigated in this study:

**Hypothesis 2 (H2).** *A consumer's hedonic value has a significant positive effect on their purchase intention regarding online food delivery platforms.*

### 2.3. Theory of Planned Behavior

Ajzen proposed the TPB in 1985, adding the "perceived behavioral control" variable to the original theory of reasoned action (TRA). Ajzen and Madden [74] conducted an empirical study in 1985 and found that TPB more closely reflected actual behavior than TRA. The primary purpose of TPB is to forecast and clarify individual behavior. It assumes that individual behavior is the result of an individual's "behavioral intention." In contrast, the individual's "attitude" (AT), "subjective norms" (SN), and "perceived behavioral control" (PBC) will affect an individual's behavioral intention. However, when consumers use online food delivery platforms, they will also be affected by some other factors. In addition, whether they used it or not, the factors have important value for platform managers regarding the improvement of the products and services on the platform. We think about the factors as External and Internal factors, so this study used TPB to discuss this issue, and the three variables are described in the following subsections.

### 2.3.1. Attitude

Ajzen [75] argued that attitude is the evaluation of the sum of the products of a person's "behavioral beliefs" and "outcome evaluations." Here, "behavioral beliefs" represent beliefs that certain results may be generated from engaging in certain behaviors. "Outcome evaluations" are an individual's evaluations of the results of said individual's behavior. That is, a stronger belief in the result of a behavior increases the likelihood that the behavior will be performed. Liang and Lim [76] found that consumers' attitudes toward buying food online have a significant positive effect on their behavioral intentions. Novela [77] pointed out that the effect of utilitarian motivation on attitude, and the attitude toward online purchase intention as well. Nyoto [78] pointed out that utilitarian value and hedonic value are important to the attitude to online shopping, and Albayrak et al. [79] also have the same result

in their study. Chang et al. [80] pointed out that utilitarian value and hedonic value are important factors in online shopping attitudes and attentions. Mosunmola et al. [81] pointed out that the attitude towards online purchases has a significant positive effect on the intention to use online purchases. In summary, we aim to explore whether consumers' attitudes toward food delivery platforms have a significant positive effect on the utilitarian value and hedonic value in this study. Therefore, hypotheses H3a and H3b are proposed:

**Hypothesis 3a (H3a).** *Consumers' attitudes have a significant positive effect on utilitarian value.*

**Hypothesis 3b (H3b).** *Consumers' attitudes have a significant positive effect on hedonic value.*

### 2.3.2. Subjective Norms

According to the study of Ajzen [75], subjective norms are determined by the perceived social pressure from other parties, such as relatives, friends, and work colleagues, to perform or not to perform an action [82,83]. Bhattacherjee [84] argued that users' acceptance intentions would be affected by subjective normative influences, which include influences from those considered important by users (e.g., family, friends). Lin et al. [60] pointed out that opinion leaders (e.g., brand ambassadors or endorsers) will affect consumers' utilitarian value and hedonic value toward products. Choi et al. [85] investigated and discussed the utilitarian value of smartphones for businesspeople; they found that subjective norms, image, and job relevance were determinants of ease of use, which is a factor in utilitarian value. Ozturk et al. [30] pointed out that subjective norms have a significant positive effect on users' continued usage intentions. Bui and Kemp [86] pointed out that consumers' attitudes, emotion regulation, and subjective norms influence repeat purchase intentions. Al-Swidi et al. [87] pointed out that subjective norms and perceived behavioral control have a significant impact on the purchase intention online. Al-Maghrabi et al. [88] explored the intention to continue purchasing products and services through online platforms. Their research found that consumers' subjective norms have a significant positive effect on hedonic value. In summary, this study intends to explore whether consumers' subjective norms toward food delivery platforms have a significant positive effect on utilitarian value and hedonic value. Therefore, hypotheses H3c and H3d are proposed:

**Hypothesis 3c (H3c).** *Consumers' subjective norms have a significant positive effect on utilitarian value.*

**Hypothesis 3d (H3d).** *Consumers' subjective norms have a significant positive effect on hedonic value.*

### 2.3.3. Perceived Behavioral Control

Ajzen [75] argued that perceived behavioral control reflects a persons' perception of the ease or difficulty of performing a given behavior. It will be subjectively affected by external factors; that is to say, individuals are be hindered by past experiences or expectations, which contain the level of understanding of self-competence (ability), the awareness of critical needs (resources), and the awareness of convenience (opportunity). In summary, perceived behavioral control means that consumers' cognition of their consumption behavior will affect their risk judgment and interests when using online food delivery platforms. Liang and Lim [76] found that consumers' past experiences when using the internet will significantly affect their behavioral intention toward online food purchases. Nguyen and Khoa [89] argued that the consumers' perceived control has a positive impact on the hedonic value in e-commerce. Ozkara et al. [70] pointed out that the perceived control have significant effect on online purchase intentions. Lee and Wu [90] pointed out that the perceived control of flow and concentration will positively affect consumers' utilitarian value. Yang et al. [91] pointed out that perceived behavioral control and subjective norms have important impacts on online shopping. In summary, the study infers that perceived behavioral control affects the utilitarian and hedonic values. Therefore, hypotheses H3e and H3f are proposed:

**Hypothesis 3e (H3e).** *Consumers' perceived behavioral control has a significant positive effect on utilitarian value.*

**Hypothesis 3f (H3f).** *Consumers' perceived behavioral control has a significant positive effect on hedonic value.*

*2.4. Mediating Effect of Utilitarian Value and Hedonic Value*

Mehmood and Hanaysha [92] explored the effect of utilitarian value and hedonic value on consumers' behavioral intentions. They found that hedonic value and utilitarian value have a mediating effect on behavioral intention. Vieira et al. [93] argued that utilitarian and hedonic value, and word-of-mouth mediate value and loyalty. Kim and Yang [94] showed that utilitarian value and hedonic value are partially mediated between representativeness heuristics and purchase intentions, and between adjustment heuristics and purchase intentions, and are fully mediated between availability heuristics and purchase intentions, and affect heuristics and purchase intentions. Afaq et al. [95] found that the hedonic value exhibited a mediating role for the relationships of re-patronage intentions with atmospheric harmony and human crowding. An and Han [96] found that hedonic value has partially mediated between conscious participation, enthusiasm and social value regarding customer satisfaction. Hashmi et al. [97] found that utilitarian value mediates website quality dimensions and the online impulse buying behavior relationship. Therefore, based on the above research, we argue that consumers' utilitarian value and hedonic value have a mediating effect on consumers' attitude, subjective norms, perceived behavioral control, and purchase intentions toward online food delivery platforms in this study. Therefore, hypotheses H4 and H5 are proposed:

**Hypothesis 4 (H4).** *Utilitarian value has a mediating effect on attitude, subjective norms, perceived behavioral control, and purchase intentions.*

**Hypothesis 5 (H5).** *Hedonic value has a mediating effect on attitude, subjective norms, perceived behavioral control, and purchase intentions.*

Based on the past research, the study used attitude, subjective norms, perceived behavioral, utilitarian value, hedonic value, and purchase intentions to discuss the consumption behaviors of food delivery platform services in Taiwan. Along with the growth of the food delivery platform service in Taiwan, it's necessary to find the key factors which affect consumers' use of the food delivery platform service.

## 3. Methodology

*3.1. Research Framework*

This study is centered on the TPB, and includes two variables—utilitarian value and hedonic value—in order to explore whether a food delivery platform user's purchase intention is affected by shopping value. A proposed research framework for verification and analysis is shown in Figure 1.

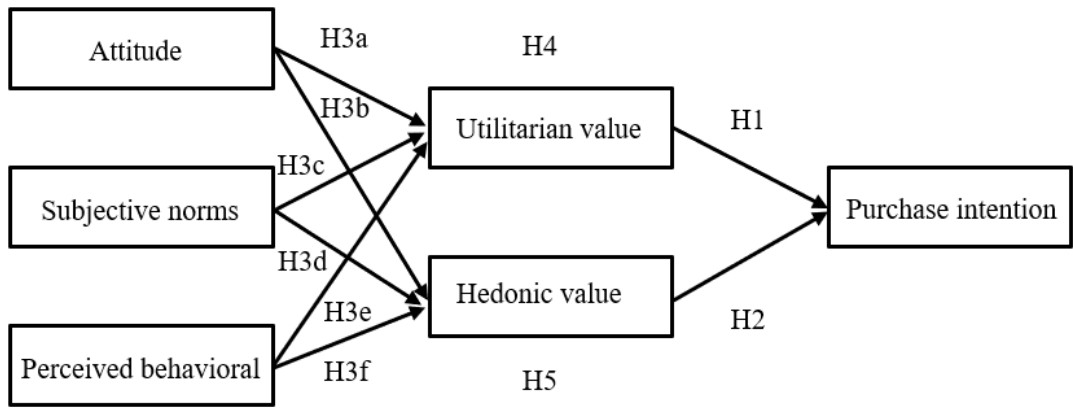

**Figure 1.** Research framework.

### 3.2. Questionnaire Design

A questionnaire was designed in order to study consumers who use food delivery platforms. The primary survey tool employed was an online questionnaire, and intentional sampling was used. People who use the service of a food delivery platform must have to connect the internet, so an online questionnaire helped us to exclude who don't use it. A structured questionnaire with a seven-point Likert scale was used. The respondents were asked to measure their responses ranging from completely disagree (1) to completely agree (7); a higher score indicates a higher evaluation of the attribute. The questionnaire items were modified by referring to the scales used in the relevant literature, with good reliability and validity. There were three parts to the questionnaire: items related to the TPB, items related to shopping value, and items requesting basic information from the respondent. The four variables proposed in the TPB—namely, attitude toward use, subjective norms, perceived behavioral control, and purchase intention—reference the works of Ajzen [75], Rogers [83], Bhattacherjee [84], Liang and Lim [76], and Lin et al. [60]. The questionnaire includes a total of sixteen items on these variables, with five items on attitudes toward use, three items on subjective norms, four items on perceived behavioral control, and four items on purchase intention. The two variables related to shopping value (utilitarian value and hedonic value) are based on the work of Batra and Ahtola [66], Crowley et al. [55], Voss et al. [67], Yeo et al. [29], and Ray et al. [57]. The questionnaire includes a total of ten items related to these two variables, with five items each related to utilitarian value and hedonic value. The third part of the questionnaire asked for basic information on the respondents, including gender, age, educational attainment, and average monthly income.

### 3.3. Methods of Data Analysis

The structural equation model (SEM) was used for the data analysis in Amos 26 and SPSS 20.0. The statistical process was comprised of a measurement model analysis and a structural model analysis.

## 4. Analysis and Results

### 4.1. Basic Sample Information

The aim of this study is to clarify the influence of purchase intention on consumers' use of food delivery platforms. Convenience sampling was used to build samples, and to evaluate the experience of using food delivery platforms. In this study, 1300 questionnaires were distributed, and after excluding invalid questionnaires, a total of 1082 valid questionnaires were received, for an 83.23% effective recovery rate. Among the recovered samples, 43% were from male respondents, while 57% were from female respondents. Regarding age, most samples were from respondents between 21 and 30 years of age (69% of the total sample size). Regarding education, the majority of the respondents had a

university-level education (52%). In terms of average monthly income, 58% of the respondents earned less than NTD 20,000 (inclusive), and 24% earned between NTD 20,001 and NTD 40,000.

## 4.2. Descriptive Statistical Analysis

This study used descriptive statistical analysis to calculate the mean and standard deviation of each variable and item, as well as to explore the potential information from respondents in the measurement of each variable. The calculated results are shown in Table 1. The mean values of the respondents' scoring on the different variables are, in descending order, perceived behavioral control (M = 5.863), utilitarian value (M = 5.727), hedonic value (M = 5.574), attitude (M = 5.502), purchase intention (M = 5.402), and subjective norms (M = 4.503). These results indicate that the respondents possess relatively higher perceived behavioral control, and feel that online food delivery platforms offer relatively more utilitarian and hedonic values. In addition, the results on subjective norms show the greatest standard deviation (SD = 1.336), meaning that the respondents had different opinions on subjective norms. The perceived behavioral control results show the least standard deviation (SD = 0.884), indicating that the respondents showed relatively high cognitive consistency with regard to perceived behavioral control.

**Table 1.** Construct reliability and validity.

| Variables | Average | Standard Deviation | Cronbach's | CR | AVE |
|---|---|---|---|---|---|
| Attitude | 5.502 | 0.870 | 0.837 | 0.877 | 0.590 |
| Subjective norms | 4.503 | 1.336 | 0.794 | 0.876 | 0.704 |
| Perceived behavioral control | 5.863 | 0.884 | 0.751 | 0.810 | 0.521 |
| Utilitarian value | 5.727 | 0.892 | 0.852 | 0.893 | 0.626 |
| Hedonic value | 5.574 | 0.929 | 0.921 | 0.940 | 0.760 |
| Purchase intention | 5.402 | 0.933 | 0.746 | 0.835 | 0.560 |

However, the answers for each item show that, regarding the statements "I think that the endorser of the online food delivery platform is very important" and "The opinions of celebrities (such as stars and Youtubers) will affect my use of food delivery platform services", 60.3% and 57.0% of responses ranged from not having an opinion to completely disagreeing with the statement, respectively. One can see that the use of endorsers and advertorial content are not key factors that increase the utilitarian value and hedonic value of consumers. Therefore, platform operators should devote more resources to the optimization of the platform itself, rather than to endorsers and advertorial costs. When it comes to the statement "Experts, scholars, news media, etc. will affect my purchase of food delivery platform services", 39.3% of responses ranged from not having an opinion to completely disagreeing with the statement. In total, 40% of the respondents indicated that they would not investigate issues affecting food delivery in the market at this stage (such as accidents to the delivery personnel and labor disputes), and this result fully reflects the fact that consumers pay more attention to their utilitarian value.

## 4.3. Measurement Model Analysis

Structural equation model (SEM) analysis was used in this study. Firstly, the reliability and validity of the questionnaire were measured as part of the measurement model analysis. The questionnaire's Cronbach's reliability was tested using SPSS 20.0, and the component reliability (CR). In terms of the convergent validity of the questionnaire, Amos 26 was used for the confirmatory factor analysis (CFA), and the average variation extraction (AVE) was calculated using a formula. Finally, the discriminant validity of the variables was tested using the square of the correlation coefficient between the AVE and each variable.

The Cronbach's values of the six variables in this study were all greater than 0.7, indicating that there was internal consistency between the corresponding variables. The CR of the potential variables is composed of the reliability of all of the observed variables, and the reliability should be greater than

0.7 [98]. Each variable has a high component reliability, which indicates that a group of observed variables measure the same potential dimension [99]. Similarly to Cronbach's reliability, CR also reflects the internal consistency between potential variables. Table 1 shows that the CR of each variable is greater than 0.7; thus, the six variables in this study have internal consistency.

The construct reliability of the questionnaire can be evaluated by its convergent and discriminant validity. AVE was used to evaluate the convergence validity, which means that the average variation in the explanatory power of the measurement variable over the potential variable should be greater than 0.5 [99]. The AVE values of all of the potential variables in Table 2 are greater than 0.50; as such, all of the variables have good convergence validity. Additionally, a good discriminant validity is demonstrated when the value of the square root of the AVE of each variable is higher than its correlation coefficient with the other variables [100]. The data in Table 2 show that each value on the diagonal is greater than all of the values to their left and bottom; thus, the value of the square root of the AVE of each variable is higher than the coefficient between the different variables. Therefore, the variables in this study have discriminant validity.

**Table 2.** Reliability analysis, and convergent and discriminant validity.

| | Attitude | Subjective Norms | Perceived Behavioral Control | Utilitarian Value | Hedonic Value |
|---|---|---|---|---|---|
| Attitude | 0.768 | | | | |
| Subjective norms | 0.461 *** | 0.839 | | | |
| Perceived behavioral control | 0.442 *** | 0.289 *** | 0.722 | | |
| Utilitarian value | 0.658 *** | 0.384 *** | 0.624 *** | 0.791 | |
| Hedonic value | 0.765 *** | 0.424 *** | 0.436 *** | 0.706 *** | 0.872 |
| Purchase intention | 0.680 *** | 0.622 *** | 0.451 *** | 0.745 *** | 0.704 *** |

Note: *** $p < 0.001$

### 4.4. Structural Model Analysis

The relationship between the different variables was explored further using path analysis, as shown in Figure 2. A goodness-of-fit test conducted on the theoretical framework yielded the following results, which lie within the acceptable limits: goodness-of-fit index (GFI) = 0.936; root mean square error of approximation (RMSEA) = 0.013; Tucker-Lewis index (TLI) = 0.976; adjusted goodness of fit index (AGFI) = 0.929; normalized fit index (NFI) = 0.952; comparative fit index (CFI) = 0.947. All of the other fit indices were above the recommended criteria.

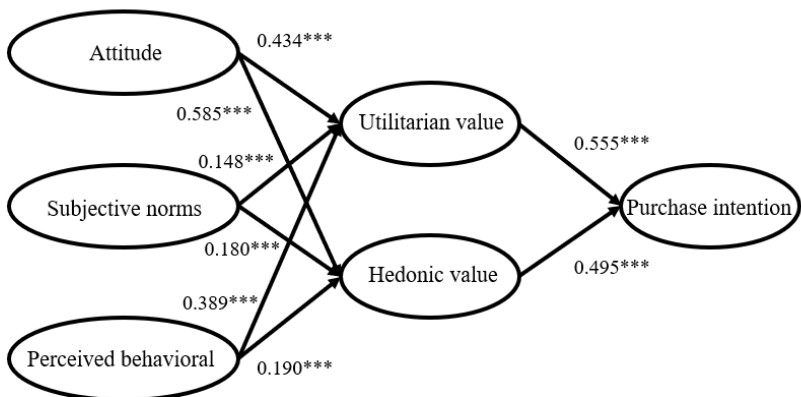

**Figure 2.** Structural pattern analysis diagram. GFI = 0.936; CFI = 0.947; NFI = 0.952; TLI = 0.976; AGFI = 0.929; RMSEA = 0.013. Note: *** $p < 0.001$.

This study adopted SEM in order to analyze the relationships among the constructs of the model. First, we examined H3a: "Consumers' attitudes have a significant positive effect on utilitarian value" ($\beta$ = 0.434, $p < 0.001$); H3b: "Consumers' attitudes have a significant positive effect on hedonic

value" ($\beta$ = 0.585, $p$ < 0.001); H3c: "Consumers' subjective norms have a significant positive effect on utilitarian value" ($\beta$ = 0.148, $p$ < 0.001); H3d: "Consumers' subjective norms have a significant positive effect on hedonic value" ($\beta$ = 0.180, $p$ < 0.001); H3e: "Consumers' perceived behavioral control has a significant positive effect on utilitarian value" ($\beta$ = 0.389, $p$ < 0.001); and H3f: "Consumers' perceived behavioral control has a significant positive effect on hedonic value" ($\beta$ = 0.190, $p$ < 0.001). We found that consumers' attitudes, subjective norms, and perceived behavioral control are significantly and positively related to utilitarian value and hedonic value. Therefore, hypotheses H3a, H3b, H3c, H3d, H3e, and H3f hold.

### 4.5. Test of the Mediating Effect of Utilitarian Value, Hedonic Value, and Purchase Intention

The third stage involved the testing of the mediating effect of utilitarian value and hedonic value on antecedent factors (attitude, subjective norms, and perceived behavioral control) and consequence factors (purchase intention). The results are as follows.

### 4.5.1. Test of the Mediating Effect of Utilitarian Value

First, by testing the mediating effect of utilitarian value on attitude and purchase intention, it was found that the effect of attitude on utilitarian value is statistically significant ($\beta$ = 0.658; $p$ < 0.001), and the effect of attitude on purchase intention is also statistically significant ($\beta$ = 0.680; $p$ < 0.001). When the mediating variable, i.e., utilitarian value, is added, the effect of attitude on purchase intention is statistically significant ($\beta$ decreased from 0.680 to 0.336; $p$ < 0.001). However, the effect of utilitarian value on purchase intention also remained significant ($\beta$ = 0.745; $p$ < 0.001). Therefore, it was concluded that utilitarian value has a partial mediating effect between attitude and purchase intention.

Second, by testing the mediating effect between subjective norms and purchase intention, the addition of utilitarian value as the mediating variable showed that the effect of subjective norms on purchase intention is statistically significant ($\beta$ = 0.393; $p$ < 0.05), while the effect of utilitarian value on purchase intention is also statistically significant ($\beta$ = 0.745; $p$ < 0.001). Thus, utilitarian value has a partial mediating effect between subjective norms and purchase intention. Finally, by testing the mediating effect between perceived behavioral control and purchase intention, the addition of the mediating variable showed that the effect of perceived behavioral control on purchase intention is statistically significant ($\beta$ = −0.022; $p$ = 0.714), but the effect of utilitarian value on purchase intention remained statistically significant ($\beta$ = 0.759; $p$ < 0.001). Therefore, utilitarian value has a full mediating effect between perceived behavioral control and purchase intention.

### 4.5.2. Test of the Mediating Effect of Hedonic Value

With the addition of hedonic value as the mediating variable in testing the mediating effect of hedonic value between attitude and purchase intention, this study used the 'Four-Step Approach' to analyze the results. The effect of attitude on purchase intention ($\beta$ = 0.343; $p$ < 0.001) and the effect of hedonic value on purchase intention ($\beta$ = 0.704; $p$ < 0.01) are statistically significant. Therefore, hedonic value has a partial mediating effect between attitude and purchase intention. When testing the mediating effect between subjective norms and purchase intention, the addition of the mediating variable showed that the effect of subjective norms on purchase intention ($\beta$ = 0.394; $p$ < 0.001) and the effect of hedonic value on purchase intention ($\beta$ = 0.704; $p$ < 0.001) are statistically significant. Therefore, hedonic value has a partial mediating effect between subjective norms and purchase intention. When testing the mediating effect between perceived behavioral control and purchase intention, the addition of the mediating variable showed that the effect of perceived behavioral control on purchase intention ($\beta$ = 0.178; $p$ < 0.001) and the effect of hedonic value on purchase intention ($\beta$ = 0.704; $p$ < 0.001) are statistically significant. Therefore, hedonic value has a partial mediating effect between perceived behavioral control and purchase intention.

## 5. Discussion

The findings of the study are consistent with the results from Choi et al. [85], Ozturk et al. [30], Bui and Kemp [86], Al-Swidi et al. [87], Al-Maghrabi et al. [88]. Customers' attitudes, subjective norms, and perceived behavioral control all have a significant and positive effect on utilitarian value and hedonic value. This finding indicates that consumers' subjective expectations, the influence of third parties, and the influence of past experiences will affect consumers' perceptions of the utilitarian value and hedonic value of online food delivery platforms. In other words, platform managers need to pay attention on the effect and efficiency of the online food delivery platform because it will positively enhance the attitude of consumer. In terms of subjective norms, platform managers need to deal with customer complaints cautiously, because third parties' reviews are a key factor which affects consumers' subjective norms. The perceived behavioral control is important, meaning that platform managers need to make platform services more easy and add more payment methods for consumers to choose. However, concerning this new type of service, Taiwanese consumers will also make a careful assessment of the service before deciding to use said service.

Regarding H1: "The utilitarian value of consumers has a significant positive effect on the purchase intention toward online food delivery platforms", the results show a significant positive correlation ($\beta$ = 0.555, $p < 0.001$). Therefore, H1 holds. This finding is consistent with the results of the research of Yeo et al. [29], Anderson et al. [58], Nejati and Moghaddam [59], Lin et al. [60], Teng and Wu [62], and Overby and Lee [50]. Consumers' utilitarian value has a significant and positive effect on their purchase intention toward online food delivery platforms, which reflects the fact that the utilitarian value of online food delivery platforms is very important for consumers, and it directly affects their purchase intention. That is to say, platform managers need to upgrade their search engines in order to allow consumers to find what they want easily and quickly.

Regarding H2: "The hedonic value of consumers has a significant positive effect on the purchase intention toward online food delivery platforms", the results show a significant positive correlation ($\beta$ = 0.495, $p < 0.001$). Therefore, H2 holds. This finding of this is consistent with the results from Ozturk et al. [30] and Ozkara et al. [70], Childers et al. [22], To et al. [26], Chiu et al. [49], Wang [71] and Ali et al. [72]. The hedonic value of online food delivery platforms is likewise valued by consumers, which shows that, although it is an online virtual platform, consumers still care about the overall shopping experience. Therefore, online food delivery platforms could provide something that can increase consumers' hedonic value; for example, the platform could add an item that can help consumers to decide today's meal.

Regarding the result of H4, it is consistent with Mehmood and Hanaysha [92], Vieira et al. [93], Kim and Yang [94], and Hashmi et al. [97]. Consumers' utilitarian value has a mediating effect between attitude, subject norms and perceived control regarding purchase intentions. Based on the results of this study, utilitarian value is so very important for its effects on consumers' purchasing decisions on food delivery platform services. Therefore, the platform managers need to pay more attention to the enhancement of the utilitarian value of their services.

Regarding the result of H5, it is consistent with Mehmood and Hanaysha [92], Vieira et al. [93], Kim and Yang [94], Afaq et al. [95], and An and Han [96]. Consumers' hedonic value has a mediating effect between attitudes, subject norms, and perceived control regarding purchase intention. Based on the results of this study, the hedonic value is very important for its effects on consumers' purchasing decisions on food delivery platform services. Thus, the platform managers need to pay more attention to the enhancement of the hedonic value.

## 6. Conclusions and Recommendations

### 6.1. Research Conclusions and Recommendations

The results presented in this paper show that consumers' attitudes, subjective norms, and perceived behavioral control have a significant and positive effect on utilitarian value and hedonic

value. Consumers' subjective expectations, the influence of third parties, and the influence of past experiences will have a significant and positive effect on their utilitarian value and hedonic value. As such, platform operators need to be cautious when creating a brand image and communicating how they solve problems in order to avoid leaving consumers with a negative impression, which will lead to the continuous spread of subsequent negative evaluations. Utilitarian value and hedonic value also have a significant and positive effect on the intention to purchase products from food delivery platforms. Regarding mediating effects, the utilitarian and hedonic values produce a mediating effect between attitude, subjective norms, and perceived behavioral control, and consumers' intentions to purchase from food delivery platforms. Moreover, utilitarian value has a full mediating effect between perceived behavioral control and purchase intentions, which indicates that factors such as ease of use, convenience, degree of enjoyment, and positive experience will all affect consumers' purchase decisions. In particular, the utilitarian value becomes more important as consumers' past experiences and anticipated future obstacles (such as payment methods or availability of internet access) are taken into consideration. Therefore, as food delivery platforms are known for their convenience, platform operators must constantly optimize the overall hardware and software quality. Meanwhile, platform operators must focus on user experience (hedonic value) in order to increase a consumers' purchase intentions.

We hope to determine whether the emergence of food delivery platforms will change the consumption patterns of consumers, as well as to use the research results as a reference for the future development of strategies in the food and beverage market. Therefore, the managers must take utilitarian value and hedonic value into consideration before making a specific program, e.g., they should design a questionnaire to confirm the satisfaction of the purchasing experience, upgrade the app system to let consumers more easily find what they need; they should also use utilitarian value and hedonic value to estimate the score of the achievement of a specific strategy. In this way, the manager can create a promotion project to increase the revenue and the use count of their platform. We also know that attitude, subjective norms and perceived behavioral control are the factors that may affect consumers, so the way in which to complete orders and deal with complaints is an essential consideration.

*6.2. Research Limitations and Future Research Directions*

While conducting experiments, we did not compare and rank service items for each delivery platform in detail, nor were the respondents asked to provide information on which food delivery platforms they use. Therefore, we hope that future research can further explore and analyze the differences among the service items offered by different brands in order to infer whether said differences will generate different research findings.

With respect to the Taiwanese food delivery market, there are still many issues to be addressed. Such issues include labor–employment relations, wage adjustments for delivery personnel, and the percentage of revenues that go to store owners, and these problems are all about finding a balance that is acceptable to consumers, delivery personnel, platform operators, and stores. Therefore, future research can include labor–employment relations, wage ratios, delivery fees, and platform commissions as research variables, with the goal of finding an acceptable balance for all parties while maximizing utility.

**Author Contributions:** Four co-authors had together contributed to the completion of this article. Formal analysis; investigation; data curation and writing—original draft preparation, H.-S.C.; investigation, data curation, review and editing, C.-H.L.; writing—original draft preparation, writing—review and editing, S.-Y.L.; and supervision, investigation, resources, H.-Y.K. All authors have read and agreed to the published version of the manuscript.

**Funding:** This research received no external funding.

**Acknowledgments:** I would like to express my sincere appreciation to all the experts who have taken the time to review this article and provide lots of valuable comments.

**Conflicts of Interest:** The author declares no conflict of interest.

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
