# Peer review of "Consumer Attitudes and Purchase Intentions toward Food Delivery Platform Services"

_sustainability, doi:10.3390/su122310177_

Round 1

Reviewer 1 Report

Research gap and research goal

 The literature review presented by the author does not show any conclusions regarding the existing research gap.

In both the Abstract and the Introduction sections, the author insufficiently specifies the purpose of the research in the theoretical context, saying only: "(...) this study is based on TBP. Utilitarian value and hedonic value are used to explore the perception that Taiwanese consumers have toward the services offered by food delivery platforms and the factors that influence their purchase decision". The nature of the extension of the TPB model with two new model constructs requires a clarification.

Additionally - in the introduction - it is necessary to explain in more detail to what context (aspect) the relationship between TPB and additional variables (utilitarian value and hedonic value) will be investigated. The author says that he hopes "to determine whether the emergence of food delivery platforms will change the consumption patterns of consumers" and "use the research results as a reference for future development of strategies in the food and beverage market". However, it is a very general and laconic description. In addition, later in the Conclusion and recommendation section, both these aspects are missing: the final naming of the mentioned changes in consumption patterns and specific proposals for the directions of strategy development on the described market. Especially the fragment concerning the recommendation and applicability of research results in business is insufficient (only two general sentences). The developing of this interesting topic would probably increase the value of the article.

Research framework

Five research hypotheses were formulated in the section Literature review and discussion. All five hypotheses are closely related to each other. Therefore, all five hypotheses should be included in the construction of the research model. The research model presents only the hypotheses from H1 to H3 and ignores the graphical presentation of the hypotheses H4 and H5 (the hypothesis describing the mediating effect of two variables: utilitarian value and hedonic value).

The structure of the text

According to the exact content of the text, the section Literature review and discussion should be titled Literature review. The section titled Discussion should be a separate part. It should follow the Analysis and results section and it should include a discussion of the results of the author's research compared to other researchers on this topic.

The text does not have a correct discussion of the results in comparison with the research of other authors. In the Analysis and results section, there are only few references to the results of other researchers, but only in the area of hypotheses from H1 to H3. The results of the verification of the H4 and H5 hypotheses were not commented on in the context of other studies.

Review of the literature

The author should expand the description of the concepts: Shopping value, Utilitarian value, Hedonic value. These concepts are relatively worse represented in the literature review presented by the author (for example, compared to the presented literature review in the field of TPB). It also has an impact on the quality of the theoretical support of research hypotheses related to these concepts.

Reviewer 2 Report

Summary:

This paper  explore the effect of factors such as attitude, subjective norms, and perceived behavioral control on purchase intention by food delivery platform users. The TPB methodology was used. I find the paper is interesting.  Overall it is well organized, however some important issues need to be addressed.

Below my detailed comments:

  • The survey is based on an Intentional sampling method that is a non-probabilistic procedure. Therefore I suggest to include a discussion on this issue in the methodological section in order to provide a measure of the bias with respect to the population of Tawian consumers (at least of the web population!). The variables gender, age, educational attainment, and average monthly income collected in the survey, for instance, can be compared with the same variables in the population.
  • A table that describes in details the single items that measure the latent constructs presented is Figure 1 is missing. I suggest to add this information.
  • In Figure 1 a suggest to change in Figure 1. THEORETICAL framework . Moreoevr, I suggest to delete the title on the top “Theory of planned behavior”
  • In table 1. I suggest to change the name of the first column. Indeed, they are the latent constructs not observed variables. Moreover, some measure reported in that Table have to be clarify. For instance, the average values how have been computed? It is the average of all the items used to measure each latent constructs? In my opinion is more correct to report the average value and the std of each observed item in each latent construct. I not agree with the title used for this table. For instance, what does it mean Scale of measure???  A correct title for the last two indicators reported is table and also for the correlation reported in table 2 would be “Reliability analysis and convergent and discriminant validity.”
  • In Figure 2 the estimated paths for the measurement model are completely missing. And there are no comments in the text of the indicators reported in title of Table 2. Moreover, it is not appropriate to report these indicators in a Table title.
  • The results reported in the sections “4.5.1. Test of the mediating effect of utilitarian value and 4.5.2. Test of the mediating effect of hedonic value” are difficult to follow and are not clear. The authors should explain what kind of models were estimated and how they could infer that a mediate effect exist by just comparing coefficients without report for example a t-test? And the values that are reported in that paragraphs are different from the estimated path values of Figure 2 where the two mediate variables are considered.

Author Response

請參考附件。

Round 2

Reviewer 1 Report

In general, the changes made by the author are small, laconic and without details (especially in the context of the cited literature, comparisons with another research/conclusions of other authors) and, in my opinion, unfortunately still insufficient.

Some of my comments/recommendations (in my opinion important for the quality of the scientific study) were not treated with due care when revising the text. Other comments resulted only in short or very generally written corrections/changes.

As below:

  1. Insufficiently specified research gap - it is not enough to write that "The other side, there are few studies use TPB, Utilitarian value and hedonic value at the same time to discuss the food delivery platform". The author should write what the research is, who described it (mention all relevant references).
  2. The more detailed description of the research goal in the theoretical context - the added content is very laconic and (note as above) do not contain any references to the literature, which is particularly important when we are talking about the theoretical context of this description.
  3. Discussion section still missing - the author only changed the section title from "Literature review and discussion" to "Literature review". However, he did not introduce a separate Discussion section in the text. It is necessary when presenting the results of own research in the scientific article. Own results should be confronted with the results of other researchers. Even if the topic is still poorly researched (this would only increase the added value of the author's research) or the topic is studied in a slightly different aspect/context/approach (then it is worth showing author’s new value/new context to the research problem). But all of this requires the preparation of a well-developed and well-documented (literature) Discussion section. So there are still only few references to the results of other researchers, but only in the area of ​​hypotheses from H1 to H3 (still in the Analysis and Results section) and three sentences added in section 4.5.2. (concerning only one variable in the model - hedonic value, nothing about utilitarian value in this context). This still does not meet the requirements of a properly conducted scientific discussion of the results obtained.

Round 3

Reviewer 1 Report

Thank you for giving me the opportunity to review this paper. The article has been partially corrected and supplemented, however I am still not fully satisfied with several aspects, namely:
- still insufficient theoretical background mainly for (1) identifying the research gap and (2) specifying the research goal and (3) formulating research hypotheses,
- a very brief Discussion section (the text was simply moved from another section and supplemented with just two new paragraphs) resulting in equally laconic Conclusions section,
- theoretical implications of the research are still difficult to find in the article,
- at the same time the formulated practical implications (largely already known to scientists and industry practitioners) are not very revealing.

Author Response

Thank you very much for your insightful comments and suggestions. We believe your comments and suggestions are appropriate and useful to us in order to improve considerably the quality of the manuscript. We have revised our paper in light of your comments and instructions.Please refer to attached document.

Round 4

Reviewer 1 Report

Thank you for the opportunity to review this article.

In my opinion the manuscript has been significantly improved.

I am asking for a final linguistic correctness check.